# The Role of Peroxiredoxin 6 in Cell Signaling

**DOI:** 10.3390/antiox7120172

**Published:** 2018-11-24

**Authors:** José A. Arevalo, José Pablo Vázquez-Medina

**Affiliations:** Department of Integrative Biology, University of California, Berkeley, CA, 94705, USA; josearevalo@berkeley.edu

**Keywords:** glutathione peroxidase, phospholipase A_2_, inflammation, lipid peroxidation, NADPH (nicotinamide adenine dinucleotide phosphate) oxidase, phospholipid hydroperoxide

## Abstract

Peroxiredoxin 6 (Prdx6, 1-cys peroxiredoxin) is a unique member of the peroxiredoxin family that, in contrast to other mammalian peroxiredoxins, lacks a resolving cysteine and uses glutathione and π glutathione S-transferase to complete its catalytic cycle. Prdx6 is also the only peroxiredoxin capable of reducing phospholipid hydroperoxides through its glutathione peroxidase (Gpx) activity. In addition to its peroxidase activity, Prdx6 expresses acidic calcium-independent phospholipase A_2_ (aiPLA_2_) and lysophosphatidylcholine acyl transferase (LPCAT) activities in separate catalytic sites. Prdx6 plays crucial roles in lung phospholipid metabolism, lipid peroxidation repair, and inflammatory signaling. Here, we review how the distinct activities of Prdx6 are regulated during physiological and pathological conditions, in addition to the role of Prdx6 in cellular signaling and disease.

## 1. Introduction

Peroxiredoxins are a ubiquitous family of highly conserved enzymes that share a catalytic mechanism in which a redox-active (peroxidatic) cysteine residue in the active site is oxidized by a peroxide [1]. In peroxiredoxins 1–5 (2-cys peroxiredoxins), the resulting sulfenic acid then reacts with another (resolving) cysteine residue, forming a disulfide that is subsequently reduced by an appropriate electron donor to complete a catalytic cycle [2,3]. In contrast to peroxiredoxins 1–5, peroxiredoxin 6 (Prdx6, 1-cys peroxiredoxin) lacks a resolving cysteine and uses glutathione (GSH) to complete its catalytic peroxidatic reaction [4,5]. Aside from the mammalian Prdx6, 1-cys peroxiredoxins that use GSH as the resolving thiol have been described in yeast [6].

Prdx6 is the only mammalian peroxiredoxin capable of reducing phospholipid hydroperoxides through its glutathione peroxidase (Gpx) activity, in addition to reducing short chain hydroperoxides [7]. Moreover, Prdx6 also expresses acidic calcium-independent phospholipase A_2_ (aiPLA_2_) and lysophosphatidylcholine acyl transferase (LPCAT) activities in separate catalytic sites [8,9]. Therefore, Prdx6 is a multi-tasking enzyme that participates in cellular signaling by modulating several pathways through its peroxidase, aiPLA_2_, and LPCAT activities. The three activities of Prdx6 are differentially regulated by the subcellular localization of the protein, substrate binding, and post-translational modifications [10]. Differential expression of the enzymatic activities of Prdx6 appears to have contrasting roles in several pathologies. Here, we review the current knowledge of the role of Prdx6 in cellular signaling. This publication is part of a forum on Prdx6 as a unique member of the peroxiredoxin family.

## 2. Regulation of the Enzymatic Activities of Prdx6

The subcellular localization of Prdx6 is crucial for the regulation of its enzymatic activities. Prdx6 exhibits maximal aiPLA_2_ activity at an acidic pH and maximal peroxidase activity at a neutral pH [10]. Hence, when Prdx6 is localized in lysosomal-type organelles such as lamellar bodies in type II pneumocytes, it functions as an aiPLA_2_, contributing to lung phospholipid metabolism [11]. In contrast, cytosolic Prdx6 functions mainly as a peroxidase under basal conditions [4]. Aside from the cytosol and lysosomal-type organelles, Prdx6 can move to the mitochondria, the plasma membrane, and possibly other organelles, after cellular stimulation with different agonists [12,13]. Prdx6 translocation to the mitochondria controls oxidative stress in the initial step of PINK1/Parkin-mediated mitophagy [14]. Prdx6 translocation to the plasma membrane has several biological effects involving its peroxidase and aiPLA_2_ activities, which will be described in detail in this manuscript. Prdx6 has also been detected in extracellular fluids, including plasma, bronchoalveolar lavage, and cerebrospinal fluid, but it is unknown whether extracellular Prdx6 functions as a peroxidase or as an aiPLA_2_ [11,15,16].

Several post-translational modifications regulate the enzymatic activities of Prdx6 by promoting changes in the intracellular localization of the protein or inactivating one or both catalytic sites. The phosphorylation of Prdx6 at threonine 177 by p38 mitogen-activated protein kinase (MAPK), after stimulation with phorbol ester, angiotensin II, or shear stress, induces a conformational change in the protein, promotes its translocation to the plasma membrane, and increases its aiPLA_2_ activity [12,17]. Prdx6 is also phosphorylated in a circadian manner, suggesting that a switch between peroxidase and aiPLA_2_ activities may occur naturally [18]. Similarly, MAPK activity is required for the targeting of Prdx6 to lysosomal-type organelles, which occurs through the binding of Prdx6 to the chaperone protein 14-3-3ε [19,20,21]. The hyperoxidation of Prdx6 at cysteine 47 inactivates the Gpx activity while increasing aiPLA_2_. In contrast to the hyperoxidation of 2-cys peroxiredoxins, the hyperoxidation of Prdx6 is irreversible [22]. The glutathionylation of Prdx6 by π GSH S-transferase (πGST) is required for the reduction of the oxidized cysteine and the completion of the peroxidatic catalytic cycle [23]. Prdx6 can also be aberrantly Sumoylated by Sumo1 at lysines 122 and 142 during oxidative stress. Aberrant sumoylation of Prdx6 reduces its gene expression and protein abundance [24,25,26].

The binding of Prdx6 to different substrates differentially regulates its GPx and aiPLA_2_ activities. At an acidic pH, when aiPLA_2_ is maximal, Prdx6 binds to reduced phospholipids. In contrast, Prdx6 binds only to oxidized phospholipids at cytosolic pH. The binding of Prdx6 to oxidized phospholipids requires the translocation of Prdx6 from the cytosol to the cell membrane, and has been observed after treating live cells with peroxides [11,27,28,29,30]. Cytosolic Prdx6 could also potentially translocate to oxidized intracellular organelle membranes, but this idea remains unexplored.

Aside from posttranslational modifications and subcellular localization, the enzymatic activities of Prdx6 can be regulated by interactions with other proteins. The direct interaction of Prdx6 with the surfactant protein A (SP-A) suppresses aiPLA_2_ in a process that is important for the regulation of lung phospholipid metabolism in lamellar bodies of type II pneumocytes [31,32,33]. Similarly, the interaction of Prdx6 with LIMP2 likely facilitates the binding of Prdx6 to phospholipids in acidic environments [34]. As described above, the binding of Prdx6 to πGST is necessary for the reduction of the oxidized cysteine and thus for the completion of the peroxidatic catalytic cycle, but not for aiPLA_2_ activity [23,35,36]. Prdx6 can also interact with the NADPH oxidase (Nox) subunits p67*^phox^* and Noxa1 [37,38]. These interactions seem to be necessary for optimal Nox activity. Interaction of phosphorylated Prdx6 with p67*^phox^* inhibits aiPLA_2_ and appears to be the mechanism that terminates Nox activation [39].

## 3. Signaling by Prdx6 Peroxidase

Peroxiredoxins are well known for their role in cellular signaling as hydrogen peroxide (H_2_O_2_) sensors. As mentioned above, Prdx6 is the only peroxiredoxin capable of reducing phospholipid hydroperoxides in addition to reducing H_2_O_2_ and short-chain peroxides. Hydroperoxide reduction occurs in three steps: peroxidation (the reduction of the hydroperoxide, which oxidizes the cysteine to sulfenic acid), resolution (the reduction of sulfenic acid), and recycling (the regeneration of the cysteine active site) [40,41]. This process occurs at the conserved cysteine 47 catalytic site and requires glutathione and πGST [5,36].

Lipid peroxidation products such as malondialdehyde (MDA) and 4-hydroxynonenal (4HNE) have traditionally been viewed as toxic byproducts implicated in the etiology of several diseases [42,43,44,45]. However, increasing evidence shows that MDA and 4HNE also elicit biological activities that result in anti-inflammatory and pro-survival effects such as Nrf2 activation and UCP3 upregulation [46,47]. Therefore, lipid hydroperoxide reduction by Prdx6 is crucial to limit peroxidation-derived pathological effects and to regulate adaptive responses mediated by electrophilic lipid peroxidation products. The only other enzyme known to be able to reduce phospholipid hydroperoxides in mammalian cells is glutathione peroxidase 4 (Gpx4) [48,49]. The expression of Prdx6 and Gpx4 likely varies among tissues, cell types, and possibly developmental stages and disease progression, but it is currently unknown if these two proteins exhibit complementary functions or if one replaces the other in particular pathophysiological conditions.

As described above, Prdx6 is recruited to peroxidized cell membranes following oxidative stress, where it reduces and hydrolyzes the oxidized sn-2 fatty acyl or the sn-2 ester (alkyl) bond of oxidized phospholipids through its GPx and aiPLA_2_ activities [30,50]. Thus, Prdx6 actively participates in the prevention and repair of lipid peroxidation. The critical role of both the GPx and aiPLA_2_ activities of Prdx6 in the repair of lipid peroxidation has been extensively demonstrated in cells, lungs, and knock-in mouse models that express either Prdx6 GPx or aiPLA_2_ activity after treatment with peroxides and exposure to paraquat or hyperoxia [28,29,51].

A novel lysophosphatidylcholine acyl transferase (LPCAT) activity of Prdx6 coupled to aiPLA_2_ was recently described [9]. Prdx6 LPCAT acylates lysophosphatidylcholine (LPC) generated by aiPLA_2_ in a continuous process, without the release of the LPC intermediate generated by aiPLA_2_. Thus, LPCAT and aiPLA_2_ combine to replace an oxidized acyl chain with palmitoyl CoA, effectively repairing oxidized cell membranes. Hence, Prdx6 possesses the ability to reduce peroxidized cell membrane phospholipids through its GPx activity and to replace the oxidized sn-2 fatty acyl group through hydrolysis/reacylation by aiPLA_2_ and LPCAT, providing a complete system for the repair of peroxidized cell membranes [30].

The recent discovery of ferroptosis, a distinct cell death pathway characterized by the intracellular accumulation of lipid hydroperoxides [52,53,54,55], is generating increasing interest in the role of lipid peroxidation in the progression of several diseases. The role of GPx4, the only other enzyme capable of reducing lipid hydroperoxides, is evident in ferroptosis [54,55]. In contrast, the role of Prdx6 as a regulator of ferroptosis has not been studied, but undoubtedly warrants further investigation.

## 4. Signaling by aiPLA_2_

Studies using Prdx6 knock-in and knock-out mice, intact lung preparations and primary cells, and an aiPLA_2_ inhibitor (MJ33) show that, aside from its role in lung phospholipid metabolism, aiPLA_2_ is necessary for the activation of NADPH oxidase 2 (Nox2) during stimulation with phorbol ester, shear stress, angiotensin II, N-formylmethionine-leucyl-phenylalanine, or endotoxin [12,37,56,57,58,59,60]. The mechanism by which Prdx6 regulates Nox2 starts with the p38 MAPK-mediated phosphorylation of Prdx6, its translocation to the plasma membrane, and the generation of LPC by aiPLA_2_. LPC is then converted to lysophosphatidic acid (LPA) by lysophospholipase D. LPA signals through LPA receptors, activating the small GTPase Rac which moves from the cytosol to the membrane to activate Nox2 [12,60,61].

Initial studies discovered a direct interaction of Prdx6 with p67*^phox^*, a cytosolic component of Nox2, and suggested that aiPLA_2_ plays a critical role in Nox2 activation [56]. Later studies showed that binding occurs only between phosphorylated Prdx6 and non-phosphorylated p67*^phox^* and that such an interaction suppresses aiPLA_2_, likely terminating Nox2 activity [39]. Contrasting evidence exists about the role of Prdx6 in the regulation of other members of the NADPH oxidase family. On the one hand, MJ33 did not inhibit angiotensin II-stimulated oxidant generation in human pulmonary smooth muscle cells, which express high levels of NADPH oxidase 1 (Nox1) [58]. On the other hand, recent evidence showed that Prdx6 binds to Noxa1, a regulator of Nox1, and that knockdown of Prdx6 or inhibition of aiPLA_2_ with MJ33 blunts Nox1-derived oxidant generation and suppresses cell migration [38].

NADPH oxidases are important activators of redox-dependent inflammatory pathways [62]. Thus, aiPLA_2_ signaling plays a role in inflammation by regulating Nox-derived oxidant generation. The inhibition of aiPLA_2_ with MJ33 protects the lungs from hyperoxia, endotoxin, and ischemia/reperfusion injury by blunting oxidant generation and preventing inflammation [58,59,63]. Our unpublished observations suggest that knock-in mice deficient in aiPLA_2_ are resistant to sepsis-induced acute lung injury. Signaling effects of aiPLA_2_, independent from the regulation of NADPH oxidases such as the generation of inflammatory mediators derived from arachidonic acid, have been proposed [64,65]. aiPLA_2_, however, does not show a preference for arachidonic acid-containing phospholipids [66,67]. Therefore, it is unclear how aiPLA_2_ could regulate inflammatory pathways derived from arachidonic acid release.

## 5. Prdx6 Signaling in Disease

### 5.1. Cancer

Peroxiredoxins have a dual role in carcinogenesis, acting as both tumor suppressors and promoters [68,69,70]. Such a dual role appears to be consistent with the established role of oxidants in both cell proliferation and death [68,71,72]. Increased Prdx6 levels have been detected in various cancers, and the role of Prdx6 in lung cancer invasion is well known [73]. aiPLA_2_ promotes cell invasion and metastasis in various models, likely through the regulation of NADPH oxidase, which is needed for cell proliferation [74]. Mice that over-express Prdx6 show a greater increase in the growth of lung tumors compared to wild-type animals. Both aiPLA_2_ and Prdx6 peroxidase activities are implicated in lung tumor development through the regulation of redox-sensitive pathways such as MAPK, JNK, JAK/STAT, and AP-1 [75,76,77].

### 5.2. Inflammation

Prdx6 appears to have a dual role in inflammatory disease. One the one hand, as described above, aiPLA_2_ is necessary for the activation of the pro-inflammatory NADPH oxidase. On the other hand, Prdx6 is crucial to counteract increased reactive oxygen species (ROS) generation and to repair oxidized cell membranes following oxidative stress. Hyperoxidized Prdx6 levels are high in the peripheral blood mononuclear cells of moderate-to-severe asthma patients, where ROS generation is also elevated [78]. Moreover, LPS-treated Prdx6-over-expressing mice show less renal apoptosis and leukocyte infiltration than wild-type controls [79]. Similarly, Prdx6 null mice show increased lung inflammation after intratracheal instillation of lipopolysaccharide (LPS) [80]. Lung inflammation and mortality induced by paraquat or hyperoxia are also increased in Prdx6 null mice [81,82]. In contrast, Prdx6 null mice show reduced inflammation, but increased hepatic oxidative damage and mitochondrial dysfunction after experimental ischemia/reperfusion [83]. Moreover, the inhibition of aiPLA_2_ with MJ33 reduces inflammation after hyperoxia or intratracheal LPS exposure [59,63]. Therefore, it is likely that Prdx6 plays a dual role in inflammatory conditions, both as an activator of inflammatory pathways related to NADPH oxidase through aiPLA_2_ and as a protective/regulatory mechanism through Prdx6 peroxidase activity.

The link between Prdx6 and NfκB, which is one of the most prominent redox-regulated pro-inflammatory regulators, has been known for some time [84]. Prdx6 protects against ischemia/reperfusion injury during liver transplantation by downregulating NfκB [85]. Similarly, Prdx6 expression is inversely correlated with NfκB during *Clonorchis sinensis* infection [86]. In contrast, extracellular Prdx6 signals through a toll-like receptor (TLR) after focal cerebral ischemia/reperfusion [87]. Other extracellular oxidized peroxiredoxins released via the exosomal pathway can bind to TLR, contributing to sustained inflammatory responses that involve NfκB signaling. The specific role of Prdx6 in this process, however, remains unknown [88]. Our unpublished observations show that the genetic inactivation (knock-in) of aiPLA_2_ prevents the nuclear translocation of NfκB in lung endothelial cells stimulated with LPS. Thus, it appears that Prdx6 peroxidase and aiPLA_2_ may play contrasting roles in NfκB signaling.

### 5.3. Metabolic Disease

Prdx6 is linked to the development of type 2 diabetes. Prdx6 null mice show low insulin and elevated blood glucose levels, compared to wild-type mice, after a glucose tolerance test. Similarly, insulin signaling and pancreatic β cell morphology are impaired in Prdx6 knockout mice. Moreover, Prdx6 null mice develop dyslipidemia [89]. Prdx6 appears to also be linked to the generation of hydroxy fatty acids, which are antidiabetic/anti-inflammatory lipid mediators synthesized in white adipose tissue via de novo lipogenesis [90]. The specific role of Prdx6 peroxidase and aiPLA_2_ activities in the development and progression of metabolic disease remains unexplored.

### 5.4. Ocular Damage

The role of Prdx6 in ocular oxidative damage has been extensively investigated [91,92]. Prdx6 expression in the eye declines with age, increasing the risk of cataract formation [26,93,94]. Moreover, Prdx6 prevents oxidative stress in human retinal pigment epithelial cells by activating the Phosphoinositide 3-kinase/Protein kinase B (PI3K/AKT) pathway [95] and blunting transforming growth factor beta (TGF-β) signaling [96]. Prdx6 is therefore necessary to prevent cataract formation and to limit aging-induced oxidative stress in the eye. The exposure of human lens epithelial cells to ultraviolet-B light results in Prdx6 hyperoxidation, increased ROS generation, and cellular toxicity [92,97]. The function of Prdx6 in the eye is the topic of a separate review article, which is part of this forum.

### 5.5. Brain Injury and Neurodegeneration 

The role of Prdx6 in neurodegenerative diseases is controversial [98]. Following a traumatic brain injury, oxidized Prdx6 is detected in the cerebrospinal fluid [16,99]. Moreover, Prdx6 signaling through TLR4 increases after an ischemic stroke, leading to NfκB-mediated inflammation and cell death [87,100,101]. Similarly, the pharmacological inhibition of aiPLA_2_ with MJ33 reduces inflammation in an experimental stroke model, supporting the idea that aiPLA_2_ has a pro-inflammatory role in the brain under such conditions [102].

The impairment of brain neurogenesis is linked to Alzheimer’s (AD) and Parkinson’s disease (PD) [103,104]. Recent work shows that Prdx6 has an inhibitory effect on neurogenesis [105]. In contrast, Prdx6 reduces oxidative stress and counteracts ROS generation, suggesting that Prdx6 plays a neuroprotective role in AD models [103]. Similarly, Prdx6 reduces inflammation and protects against disruption to the blood–brain barrier in a model of multiple sclerosis [106]. Prdx6 hyperoxidation is observed in a PD mouse model [104]. As discussed above, the hyperoxidation of Prdx6 is irreversible, and inactivates the Gpx function. Therefore, the dual role of Prdx6 in neurodegeneration seems to be linked to the differential expression of Prdx6 peroxidase and aiPLA_2_ activities.

### 5.6. Male Infertility

Prdx6 plays a crucial role in spermatozoon fertilizing ability [107,108]. Lower levels of Prdx6 are associated with impaired sperm function and poor DNA integrity in infertile men [109]. Sperm motility, viability, fertilization, and blastocyst rates are lower in Prdx6 null mice than they are in wild-type controls [110]. The role of Prdx6 in male fertility is extensively discussed in a separate review article, which is part of this forum.

## 6. Conclusions and Future Directions

Prdx6 is an intriguing enzyme that contributes to the regulation of a plethora of signaling pathways related to its diverse catalytic activities. The contrasting biological roles of Prdx6 are likely explained by the delicate differential expression of its peroxidase and aiPLA_2_ activities. Further investigations on the role of Prdx6 in cell signaling should focus on understanding the mechanisms that control the shift between Prdx6 peroxidase and aiPLA_2_ activities under physiological and pathological conditions. Similarly, the role of extracellular Prdx6 in inflammatory signaling remains underexplored. Moreover, the role of Prdx6 as a complete lipid peroxidation repair enzyme warrants further studies to elucidate how Prdx6 regulates cellular signaling by lipid peroxidation products, as well as the possible role of Prdx6 in novel pathways of cell death.

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
