# Peer review of "The Role of Peroxiredoxin 6 in Cell Signaling"

_antioxidants, 2018, doi:10.3390/antiox7120172_

Reviewer 1 Report

This review is supported by the acknowledged pioneering expertise of one of the authors in Prdx6. It is a good compilation of the state of the art regarding this intriguing and interesting protein and is appropriate to be part of the Prdx6 forum. Several minor amendments should have to be done:

- Lines 27-29: 1-Cys peroxiredoxins have also been described in other organisms (yeast) that use GSH as the resolving thiol. This should be mentioned.

- Lines 100-107: the review by Wood et al (2003) in TIBS is more appropriate as a reference for the catalytic cycle of peroxiredoxins.

- Lines 117-119: a reference is lacking to back the mentioning of Gpx4.

- Line 226: A concluding, end of paragraph, reflection on the "obscure" relationships between Prdx6 and NFkB would be appreciated similar to final comments in other paragraphs along the review, to add author’s judgement.

- Line 260: “role” instead of “roll”.

- Lines 262-263: Reference 105 is not consistent with the citation related to Parkinson disease mouse model nor hyperoxidation of Prdx6.

Author Response

We want to thank the reviewer for the constructive comments. The MS has been modified following the reviewer’s suggestions. All modifications are marked in red in the current version of the MS.

Response to critics:

This review is supported by the acknowledged pioneering expertise of one of the authors in Prdx6. It is a good compilation of the state of the art regarding this intriguing and interesting protein and is appropriate to be part of the Prdx6 forum. Several minor amendments should have to be done:

- Lines 27-29: 1-Cys peroxiredoxins have also been described in other organisms (yeast) that use GSH as the resolving thiol. This should be mentioned.

 We have added the following sentence and reference as suggested by the reviewer (Lines 29-31):

Aside from the mammalian Prdx6, 1-cys peroxiredoxins that use GSH as the resolving thiol have been described in yeast [6].

6. Pedrajas, J.R.; McDonagh, B.; Hernandez-Torres, F.; Miranda-Vizuete, A.; Gonzalez-Ojeda, R.; Martinez-Galisteo, E.; Padilla, C.A.; Barcena, J.A. Glutathione Is the Resolving Thiol for Thioredoxin Peroxidase Activity of 1-Cys Peroxiredoxin Without Being Consumed During the Catalytic Cycle. Antioxid Redox Signal 2016, 24, 115-128, DOI 10.1089/ars.2015.6366 [doi].

- Lines 100-107: the review by Wood et al (2003) in TIBS is more appropriate as a reference for the catalytic cycle of peroxiredoxins.

We have added the reference as suggested (reference 40 in the current version, lines 106-109):

 Hydroperoxide reduction occurs in three steps: peroxidation (reduction of the hydroperoxide which oxidizes the cysteine to sulfenic acid), resolution (reduction of sulfenic acid), and recycling (regeneration of the cysteine active site) [40,41].

          40. Wood, Z.A.; Schroder, E.; Robin Harris, J.; Poole, L.B. Structure, mechanism and regulation of peroxiredoxins. Trends Biochem Sci 2003, 28, 32-40, DOI S0968-0004(02)00003-8 [pii].

- Lines 117-119: a reference is lacking to back the mentioning of Gpx4.

 The following reference has been added (48,49 in line 121):

48. Imai, H.; Nakagawa, Y. Biological significance of phospholipid hydroperoxide glutathione peroxidase (PHGPx, GPx4) in mammalian cells. Free Radic Biol Med 2003, 34, 145-169, DOI S0891584902011978 [pii].

         49. Scheerer, P.; Borchert, A.; Krauss, N.; Wessner, H.; Gerth, C.; Hohne, W.; Kuhn, H. Structural basis for catalytic activity and enzyme polymerization of phospholipid hydroperoxide glutathione peroxidase-4 (GPx4). Biochemistry 2007, 46, 9041-9049, DOI 10.1021/bi700840d [doi].

- Line 226: A concluding, end of paragraph, reflection on the "obscure" relationships between Prdx6 and NFkB would be appreciated similar to final comments in other paragraphs along the review, to add author’s judgement.

 The following text has been added (lines 231-238):

Other extracellular oxidized peroxiredoxins released via the exosomal pathway can bind to TLR, contributing to sustained inflammatory responses that involve NfκB signaling. The specific role of Prdx6 in this process, however, remains unknown [88]. Our unpublished observations show that genetic inactivation (knock-in) of aiPLA2 prevents nuclear translocation of NfκB in lung endothelial cells stimulated with LPS. Thus, it appears that Prdx6 peroxidase and aiPLA2 may play contrasting roles in NfκB signaling.

- Line 260: “role” instead of “roll”.

The typo has been corrected

- Lines 262-263: Reference 105 is not consistent with the citation related to Parkinson disease mouse model nor hyperoxidation of Prdx6

An appropriate reference (now reference 104) has been added:

104. Yun, H.M.; Choi, D.Y.; Oh, K.W.; Hong, J.T. PRDX6 Exacerbates Dopaminergic Neurodegeneration in a MPTP Mouse Model of Parkinson's Disease. Mol Neurobiol 2015, 52, 422-431, DOI 10.1007/s12035-014-8885-4 [doi].

Reviewer 2 Report

It was very nice to read this short review about Prx6 snd its role in the cell! 

The topic is undoubtedly interesting and well treated also for non expert readers. It is well written, references are updated, the length is adequate and the topics are nicely organized. Due to the importance of this enzyme in the antioxidant defense system, I recommend publication in this journal. 

Author Response

We want to thank the reviewer for reading our MS. We agree with the reviewer that the topic is quite interesting.